# Recent Molecular Characterization of Porcine Rotaviruses Detected in China and Their Phylogenetic Relationships with Human Rotaviruses

**DOI:** 10.3390/v16030453

**Published:** 2024-03-14

**Authors:** Mengli Qiao, Meizhen Li, Yang Li, Zewei Wang, Zhiqiang Hu, Jie Qing, Jiapei Huang, Junping Jiang, Yaqin Jiang, Jinyong Zhang, Chunliu Gao, Chen Yang, Xiaowen Li, Bin Zhou

**Affiliations:** 1MOE Joint International Research Laboratory of Animal Health and Food Safety, College of Veterinary Medicine, Nanjing Agricultural University, Nanjing 210014, China; qiaoml2024@163.com (M.Q.); 15005658994@163.com (M.L.); 2Shandong Engineering Research Center of Pig and Poultry Health Breeding and Important Disease Purification, Shandong New Hope Liuhe Co., Ltd., Qingdao 266000, China; youglion@163.com (Y.L.); qingjie94@163.com (J.Q.); sicaujiapeihuang@163.com (J.H.); zhangjy3@newhope.cn (J.Z.); gaochunliu1@newhope.cn (C.G.); 18864808615@163.com (C.Y.); 3Beef Cattle Industry Development Center, Fangshan 033100, China; 18404982338@163.com; 4College of Animal Science, Xichang University, Xichang 615012, China; zhiqianghu0624@163.com; 5China Agriculture Research System-Yangling Comprehensive Test Station, Xianyang 712100, China; jiangjunping1@newhope.cn (J.J.); jiangyq@newhope.cn (Y.J.)

**Keywords:** porcine rotavirus A (RVA), serotypes, molecular characteristics, prevalence, cross-species transmission

## Abstract

Porcine rotavirus A (PoRVA) is an enteric pathogen capable of causing severe diarrhea in suckling piglets. Investigating the prevalence and molecular characteristics of PoRVA in the world, including China, is of significance for disease prevention. In 2022, a total of 25,768 samples were collected from 230 farms across China, undergoing porcine RVA positivity testing. The results showed that 86.52% of the pig farms tested positive for porcine RVA, with an overall positive rate of 51.15%. Through the genetic evolution analysis of VP7, VP4 and VP6 genes, it was revealed that G9 is the predominant genotype within the VP7 segment, constituting 56.55%. VP4 genotypes were identified as P[13] (42.22%), P[23] (25.56%) and P[7] (22.22%). VP6 exhibited only two genotypes, namely I5 (88.81%) and I1 (11.19%). The prevailing genotype combination for RVA was determined as G9P[23]I5. Additionally, some RVA strains demonstrated significant homology between VP7, VP4 and VP6 genes and human RV strains, indicating the potential for human RV infection in pigs. Based on complete genome sequencing analysis, a special PoRVA strain, CHN/SD/LYXH2/2022/G4P[6]I1, had high homology with human RV strains, revealing genetic reassortment between human and porcine RV strains in vivo. Our data indicate the high prevalence, major genotypes, and cross-species transmission of porcine RVA in China. Therefore, the continuous monitoring of porcine RVA prevalence is essential, providing valuable insights for virus prevention and control, and supporting the development of candidate vaccines against porcine RVA.

## 1. Introduction

Rotaviruses (RVs), which belong to the *Reoviridae* family, are a significant cause of diarrhea in children and young animals globally [1,2]. RVs were identified more than 60 years ago in rectal swabs from monkeys and intestinal biopsies from mice [3]. Soon after, in 1973, human RV was discovered in duodenal biopsies from nine children [4]. Rotavirus A (RVA) infects humans and animals and is the most common subtype that causes diarrhea in sucking piglets. It accounts for more than 90% of diarrhea cases caused by RV in commercial pig populations [5,6]. The World Health Organization (WHO) estimates that RV causes approximately 450,000 deaths each year, with more than 90% of these deaths occurring in developing countries in Asia and Africa [3]. Porcine RV infection is a common enteric infectious disease with watery diarrhea, vomiting, anorexia, and dehydration as the main clinical features [7]. Current studies have suggested that porcine RVs may be the pathogen of RV infection in humans, dogs and other species [8,9]. Patterns of RV genotypes have evolved through interspecies transmission and recombination events [10].

The mature RV particle encapsulates a genome of 11 segments of double-stranded (ds) RNA encoding six structural proteins (VP1 to VP4, VP6 and VP7) and five or six non-structural viral proteins (NSP1 to NSP5/6) [5]. RVs are classified into ten groups (A–J) based on the antigenic relationships of the VP6 protein, which can stimulate the body to produce IgA and determine the specificity of the RV serotype [11,12]. The structural proteins VP7 and VP4, which determine the G and P genotypes of RV, together form the outer capsid of RV, and are important antigens that neutralize and induce the production of neutralizing antibodies [13]. Porcine RVA (PoRVA) encompasses a diverse array of genotypes, including G, P and I genotypes. Twelve G genotypes (G1 to G6, G8 to G12 and G26), fifteen P genotypes (P[1] to P[8], P[11], P[13], P[19], P[23] and P[26] to P[28]) and four I genotypes (I1, I2, I5 and I14) of porcine RVA have been reported worldwide [14,15,16]. G genotypes (i.e., G3 to G5, G9 and G11) are often freely combined with P genotypes (i.e., P[5] to P[7], P[13] and P[28]). G5P[7], G4P[6] and G4P[7] were the most common genotype combinations worldwide. Few studies have focused on the genotype of VP6 in pigs, resulting in limited reports on the combination of dominant G/P/I genotypes [5].

The diversity of RV strains poses significant challenges for vaccine development [10]. The trivalent live attenuated vaccine, based on the pandemic G/P combination, was found to be safe and effective in protecting piglets against diarrhea caused by homologous virulent strains. However, bivalent vaccines (containing G5P[7] and G9P[7] strains) were not effective in preventing infections caused by G8P[1], G9P[23] and G8P[7] RVA strains [17]. Therefore, understanding genetic diversity is essential for the development, optimization and improvement of vaccines, as there is limited cross-protection against heterologous strains. Therefore, in this study, we reported the prevalence of RVA in fecal samples collected from 23 provinces in China in 2022, and analyzed the proportions of genotypes in different regions to understand the prevalence characteristics of RVA in China, the largest pig-farming country.

## 2. Materials and Methods

### 2.1. Sample Preparation

In 2022, a total of 25,768 fecal samples from sucking piglets with diarrhea were collected from 230 different industrial farms in 23 provinces across China. These samples were diluted with two volumes of cold phosphate-buffered saline (PBS) and purified by centrifugation at 5000× *g* for 1 min. A 300 μL aliquot of supernatant was extracted from each sample, and the total RNA was then extracted using the Virus DNA/RNA Extraction Kit from Bioer (Hangzhou, China), following the manufacturer’s instructions.

### 2.2. Reverse Transcription-Quantitative PCR (RT-qPCR)

To determine the viral load in each sample, RT-qPCR was performed by using the TransScript Probe One-Step qRT-PCR Kit (TransGen, Beijing, China) on a Step One Plus instrument (ABI). Briefly, the procedure involved an initial reverse transcription at 45 °C for 5 min, followed by pre-denaturation at 94 °C for 30 s. The qPCR reactions were run for 40 cycles with denaturation at 94 °C for 5 s, annealing and extension at 60 °C for 30 s. The sample was considered positive for PoRVA if the CT value was less than 35 or between 35 and 40 in two repeated assays. The farm was considered positive for PoRVA if one or more diarrheal samples tested positive for PoRVA. The primers and probes utilized for the detection of PoRVA nucleic acid were specifically designed to target the conserved region of the NSP3 gene, as outlined in Appendix A.

### 2.3. Amplification of PoRVA Genes

Specific primers targeting the VP1 to VP4, VP6, VP7 and NSP1 to NSP5 genes of PoRVA were designed based on the conserved regions (Appendix A) and synthesized by Sangon Bioengineering Co., Ltd. (Shanghai, China). The reverse transcription and amplification of the selected RNAs were performed by using the HiScript II One-Step RT-PCR Kit (Dye Plus) following the manufacturer’s instructions (Vazyme, Nanjing, China). The standard program was below: initial reverse transcription at 45 °C for 25 min, pre-denaturation at 94 °C for 5 min, 32 cycles of denaturation at 94 °C for 30 s, annealing at 55 °C for 30 s, extension at 72 °C for 1, 2, or 1.5 min (for VP7, VP4 and VP6 genes, respectively) and final extension at 72 °C for 8 min. All RT-PCR products were subjected to gel electrophoresis on a 1.5% agarose gel, and the target bands were finally verified using a UV transilluminator. The fragments of VP7, VP4 and VP6 genes were cloned into the pMD-18T vector (TaKaRa, Beijing, China) followed by successful transformation into *E. coli* DH5α competent cells. The plasmids were sequenced using the Sanger approach by Sangon Bioengineering Co., Ltd.

### 2.4. Genotyping and Phylogenetic Analysis

The genotypes of individual genes of the study strains were determined with the Virus Pathogen Resource (ViPR) automated genotyping tool (https://www.viprbrc.org/brc/rvaGenotyper.spg?method=ShowCleanInputPage&decorator=reo, accessed on 17 June 2023). The sequence similarities were analyzed by using BLAST (http://blast.ncbi.nlm.nih.gov/Blast.cgi, accessed on 12 April 2023). The complete sequences of the RVA VP7, VP6 and VP4 genes, along with the complete genome-wide sequences of the LYXH2, were utilized to construct a phylogenetic tree. Reference sequences downloaded from GenBank were also incorporated in the analysis. The neighbor-joining method with 1000 bootstrap replicates for each gene was applied using MEGA 11 software. The initial tree was drawn to scale, with branch lengths representing the number of substitutions per site. Visualization was conducted using iTOL v6 (Interactive Tree of Life, http://itol.embl.de/, accessed on 28 February 2024). To assess the genomic characteristics of the LYXH2, its nucleotide sequences were compared to the complete genome sequences of rotavirus A (RVA) present in the GenBank database.

### 2.5. Nucleotide Sequence Accession Numbers

The nucleotide sequence data obtained in this study have been uploaded to the GenBank database. The accession numbers for VP7, VP4 and VP6 gene sequences are OQ743847-OQ743991, OQ799656-OQ799745 and OQ799746-OQ799879, respectively (Appendix A). The accession numbers of the CHN/SD/LYXH2/2022/G4P6I1 genome (except for VP4, VP6 and VP7) are OQ799880-OQ799887.

## 3. Results

### 3.1. Prevalence of PoRVA in China, 2022

In 2022, a total of 25,768 diarrhea samples were collected from 230 pig farms in 23 provinces across China and tested for PoRVA by qRT-PCR. The results showed that the prevalence among provinces significantly varied with positive rates ranging from 29.94% to 87.10%. Additionally, the average frequency of positive samples was found to be 51.15%. More than half of the samples tested positive for PoRVA in 52.17% (12/23) of the provinces. Furthermore, the prevalence of positive farms ranged from 70% to 100% in all of the tested provinces. Moreover, in 82.61% (19/23) of the provinces, more than 80% of the farms tested positive for PoRVA. The average prevalence rate of PoRVA at the farm level in China in 2022 was 86.09% (Table 1).

To analyze the patterns of PoRVA prevalence, the pig farms were categorized into five groups based on their geographical location: northern, central, eastern, southern and southwestern China (Figure 1). The positive rates of PoRVA were higher in central China, both in samples (59.87%) and farms (91.84%), compared to other regions. The prevalence of PoRVA in eastern China was relatively lower than in the other four regions, as indicated by the positive rate at the sample and farm levels (Table 1).

### 3.2. Genotypes and Distribution of PoRVA

To elucidate the genetic characteristics of PoRVA circulating in China, the VP7, VP4 and VP6 genes of 180 positive samples (CT value less than 30) from different farms were amplified using one-step RT-PCR. Full-length VP7 (*n* = 145), VP4 (*n* = 90) and VP6 (*n* = 134) genes were sequenced, followed by phylogenetic analysis presented in Figure 2a, Figure 3a and Figure 4a. The VP7 gene sequences were clustered into nine branches: G9 (56.55%), G5 (14.48%), G1 (8.97%), G26 (6.90%), G4 (6.21%), G3 (3.45%), G12 (2.07%), G11 (0.69%) and G2 (0.69%) (Figure 2a,c). The P genotypes were clustered into five branches, including P[13] (42.22%), P[23] (25.56%), P[7] (22.22%), P[6] (8.89%) and P[3] (1.11%) (Figure 3a,c). Only two I-genotypes (I5 and I1) were identified with proportions of 88.81% and 11.19% (Figure 4a,c).

### 3.3. Distribution of PoRVA Genotypes in Different Regions in China

The G9 genotype was predominant in all five regions, with proportions ranging from 50.00% to 64.71%. The G5 genotype was also detected in five regions, but its prevalence was significantly lower than that of G9. The G1, G4 and G26 genotypes showed regional distribution in China. G1 was not detected in northern China, and G4 and G26 were not found in central China (Figure 2b). P[13] was the major genotype in all regions except central China, where P[23] was the dominant genotype. The proportion of P[13] and P[23], along with P[7], was 90%. These strains were circulating in all five regions of China. The other two genotypes of VP4, P[6] and P[3], were found in three regions and one region of China, respectively (Figure 3b). The I5 genotype was the dominant genotype in all of the regions, ranging from 80.65% to 100%. The I1 genotype of PoRVA was only circulating in southern, eastern, and southwestern China (Figure 4b).

### 3.4. Main Genotype Combinations of VP7, VP4 and VP6 Genes

The combinations of the VP7, VP4 and VP6 genes of PoRVA were obtained, and 79 samples were detected for the G/P/I genotypes. G9P[23]I5 (22.78%) was the most common combination of PoRVA, followed by G9P[13]I5 (15.19%), G5P[13]I5 (13.92%) and G9P[7]I5 (10.13%) (Table 2). The results demonstrated that the combinations of VP7 and VP4 genes exhibited significant diversity. Nonetheless, it was noticeable that the majority of these combinations were linked with the I5 genotype of VP6.

### 3.5. Homology Analysis of PoRVA in China, 2022

The highest sequence similarity of the VP7, VP4 and VP6 genes obtained in this study was submitted to the standard nucleotide BLAST program on NCBI. As shown in Appendix A, approximately 73.79% of the VP7 sequences, 93.33% of the VP4 sequences and 78.36% of the VP6 sequences were closely related to the PoRVAs reported previously. Interestingly, it was found that 18.62% of VP7 sequences, 6.67% of VP4 sequences and 21.64% of VP6 sequences were similar to human RVA sequences. All three genes of two strains (G4P[6]I1) from Shandong and one strain (G9[P6]I1) from Guizhou showed high homology with human RVA strains instead of PoRVA strains. Among the analyzed PoRVA strains, eight strains exhibited the highest similarity to the VP7 gene of the strains isolated from giant pandas. Additionally, three RVA strains demonstrated the highest homology of VP7 genes with the strains isolated from dogs. These data indicate the possibility of interspecific transmission.

### 3.6. Complete Genomic Analysis of the Special PoRVA Strain

We attempted to sequence the complete genomes of three RVA strains that were suspected to have originated in humans. These strains displayed the greatest homology with human RVA strains, specifically in relation to the VP7, VP4 and VP6 genes. However, only the G4P[6]I1 strain from Shandong province was successfully sequenced. This strain was designated as CHN/SD/LYXH2/2022/G4P6I1 (LYXH2 for short), and its genotypes were identified as G4-P[6]-I1-R1-C1-M1-A8-N1-T1-E1-H1 (Figure 5).

Phylogenetic analyses were conducted between the closely related reference RVA strains of the same genotype. The sequence most closely related to the VP7 gene fragment of LYXH2 was identified from the human RVA strain RVA/human-WT/CHN/SZ18-2049/2018/G4P[6] (Figure 5a). Homology analysis revealed that the strain with the highest similarity to the VP7 fragment was also SZ18-2049 (Appendix A). The strain most closely associated with the genetic evolution of VP4 and NSP1 fragments was RVA/Human-wt/CHN/R1954/2013/G4P[6] (Figure 5b,g), which was documented as a porcine-like human strain in 2015 [18]. The VP6 gene fragment exhibited a high degree of similarity to the R946 strain (Figure 5c), showing 100% amino acid homology (Appendix A). The RVA/Human-wt/CHN/R946/2006/G3P[6] strain was identified in 2015 as a potential recombinant, incorporating a human RVA-like gene fragment into the genetic background of the porcine RVA strain [18]. Gene fragments VP1, VP2, NSP2 and NSP5 shared a close genetic relationship with RVA/Human-wt/CHN/LL3354/2000/G5P[6] (Figure 5d,e,h,k), an isolate obtained from the fecal samples of children under 5 years old suffering from diarrhea in China [19]. The remaining genes showed close relationship to porcine strains (Appendix A).

Remarkably, seven genes of the LYXH2 genome, including VP1, VP4, VP6, VP7, NSP2, NSP3 and NSP5, showed a high degree of homology with human RVA strains rather than other previously-reported PoRVA strains (Table 3). These data provide evidence for interspecies transmission and reassortment events between porcine and human RVAs.

## 4. Discussion

RV is a zoonotic pathogen that causes severe diarrhea in young animals, weakening their immune system and making them susceptible to other pathogens, which results in substantial economic losses in the pig industry [20]. Epidemiological survey data have shown that PoRVA is endemic and widespread in large-scale pig farms worldwide [15,16]. Until now, there has been no nationwide epidemiological investigation specifically targeting multiple genes of PoRVA in China, except a few studies that reported the positive rate of pig samples and farms in individual provinces [21,22,23]. For instance, a study conducted from 2013 to 2019 reported an RVA positive rate of 7% in Zhejiang, Shandong, Jiangsu and Shanxi provinces [24]. Another study conducted from 2017 to 2019 found an RVA positive rate of 16.83% (100/394) in East China [23]. In 2022, a study conducted in Heilongjiang province reported an RVA positive rate of 4.3% (12/280) [25]. Additionally, a study conducted in Jiangsu province in 2022 reported an RVA positive rate of 12.5% (11/88) [26]. For a comprehensive understanding of the prevalence of PoRVA in China, a large number of porcine diarrheal samples were collected from the major provinces in pig production. Our study showed that the positive rates of the samples and farms in 2022 were 51.70% and 86.26%, suggesting that PoRVA infections are widespread in China (Table 1). The high positive rates observed further emphasize the significance of addressing the potential harm caused by rotavirus infection, highlighting the importance of further research and intervention. The significant increase in the prevalence of RVA in recent years can be attributed to various factors. The complexity of pig diseases, the constant turnover of pig herds resulting in varying levels of neutralizing antibodies in sows and the increasingly complex overall disease burden on pig farms have all contributed to the rise in morbidity and high viral load in the environment. This makes it challenging to completely eliminate RV viral nucleic acids with disinfectants, and even after disinfection, the virus persists, leading to piglet infection due to viral enrichment. Additionally, existing inactivated RVA vaccines have shown limited efficacy in boosting neutralizing antibody levels in sows. Cross-protection limitations also exist between different RVA genotypes, rendering vaccines targeting specific genotypes ineffective against others. This scenario could potentially drive the emergence of new genotypes. Furthermore, the increasing occurrence of rotavirus recombination in recent years can also contribute to higher infection rates [2,27,28].

RVs are always distinguished by different G, P and I genotypes based on VP7, VP4 and VP6 genes. Due to the segmented genome, different genotypes of RVs tend to form random combinations. G9 as an emerging genotype in pigs and humans worldwide is often associated with P[7], P[13], P[19] and P[23] [29]. G9P[23] was the dominant genotype combination in Germany, Japan and Korea [13,30,31]. Interestingly, we found that the most prevalent genotypes for VP7, VP4 and VP6 in China were G9, P[13] and I5. In addition, the dominant genotype combination was G9P[23]I5, followed by G9P[13]I5 and G5P[13]I5 (Table 2). The dominance of the G9 and I5 genotypes was evident in all five regions of China, while the P[13] genotype was prominent, but not the sole leader. It has been reported that the potential for cross-protection between different genotypes of RV strains is limited [32]. Given the constant evolution of rotavirus genotypes [5], it is imperative to develop vaccines based on the prevailing epidemic genotypes. Therefore, our results would contribute to understanding the prevalence of RVA in pigs in China and to develop novel vaccines against the dominant genotypes. In addition, only two I genotypes were detected in this study, suggesting that universal methods could be developed to target VP6 nucleic acids or antibodies for assessing RV infection in pigs. The success rates of sequencing targeting the VP7, VP4 and VP6 genes were 80.56%, 50.00% and 74.44%, respectively. More conservative and genotype-specific primers should be developed in future studies.

Previous studies have shown that certain PoRVA strains are highly similar to human RVAs and often experience interspecies recombination and reassortment events [8,33]. The existence of interspecific transmission and recombination between pigs and humans could be demonstrated by conducting whole genome sequencing of RVs [8,18,27]. In line with this evidence, the homology analysis of our data showed that one or more genes of certain strains exhibited a higher homology with human RVA strains rather than porcine RVA strains. Furthermore, based on the whole genome of the LYXH2 strain from a piglet with RVA-induced diarrhea, seven out of eleven genes showed high homology to human RVA strains, but not to PoRVA strains. In addition, genetic evolutionary analysis indicated that several gene segments of LYXH2 shared close relationships with human strains R1954, R946 and LL3354, all of which have been identified as recombinant strains originating from both humans and pigs [18,19]. This phenomenon suggests instances where pig strains reinfect pigs following recombination with human strains, underscoring the dynamic interplay between human and porcine rotaviruses. These data suggest that during the long evolutionary process, human and porcine RVAs have continuously crossed species barriers and frequently recombined with each other. Despite interspecific barriers and limitations in host range, interspecific transmission plays a significant role in the diversity and evolution of rotaviruses. Pigs have served as hosts for potential zoonotic disease transmission and the emergence of new genotypes [2,27,28]. Future research should focus on the virulence and transmissibility of recombinant strains between humans and pigs.

We conducted an epidemiological survey of probable RVA-positive samples collected from 23 provinces in 2022. The results showed that PoRVA was widespread in different regions of China, with a high prevalence in both samples and farms. The dominant genotypes were G9, P[13] and I5. The most prevalent genotype combination was G9P[23]I5. Furthermore, interspecies transmission of RVA and complex reassortment involving human and porcine RVAs were demonstrated. Monitoring the epidemiology of PoRVA can provide insights for the development of vaccines and other preventive measures.

## Figures and Tables

**Figure 1 viruses-16-00453-f001:**
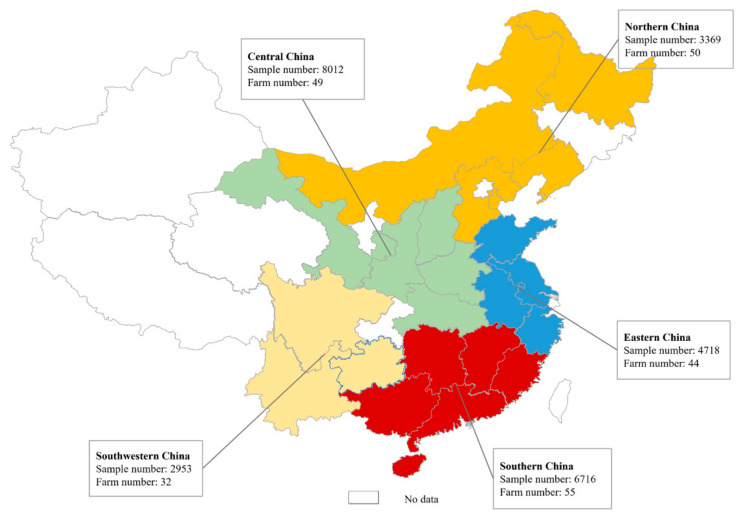
Numbers of collected samples and farms in each of the geographical divisions of China defined in this study. Northern China included Hebei, Heilongjiang, Liaoning, Inner Mongolia and Tianjin. Central China included Gansu, Henan, Shanxi, Shaanxi and Hubei. Eastern China including Anhui, Shandong, Jiangsu and Zhejiang. Southern China included Fujian, Guangdong, Guangxi, Hainan, Hunan and Jiangxi. Southwestern China included Guizhou, Sichuan and Yunnan.

**Figure 2 viruses-16-00453-f002:**
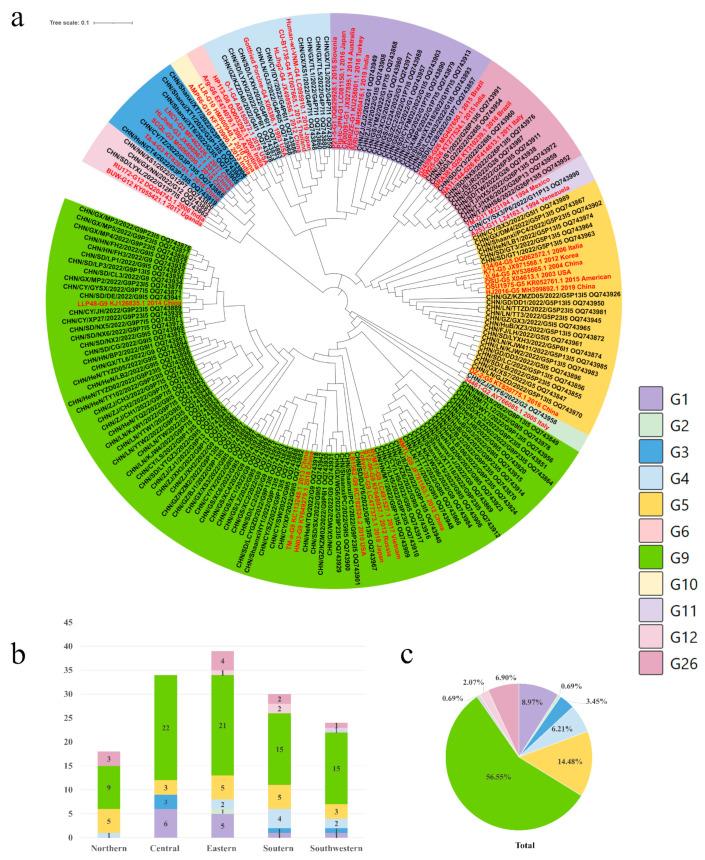
(**a**) The genotype distribution and genetic and phylogenetic analysis of the VP7 gene in different regions from diarrheal pigs. The trees of the VP7 gene were created via neighbor-joining analysis using MEGA 11 software with 1000 bootstrap replicates. Sequences of the various G genotypes of reference RVA strains were obtained from GenBank. The reference sequence was shown in red. (**b**) The genotype proportions of the VP7 gene at the farm level in different regions in 2022. (**c**) Proportional distribution of genotypes in the VP7 gene in 2022.

**Figure 3 viruses-16-00453-f003:**
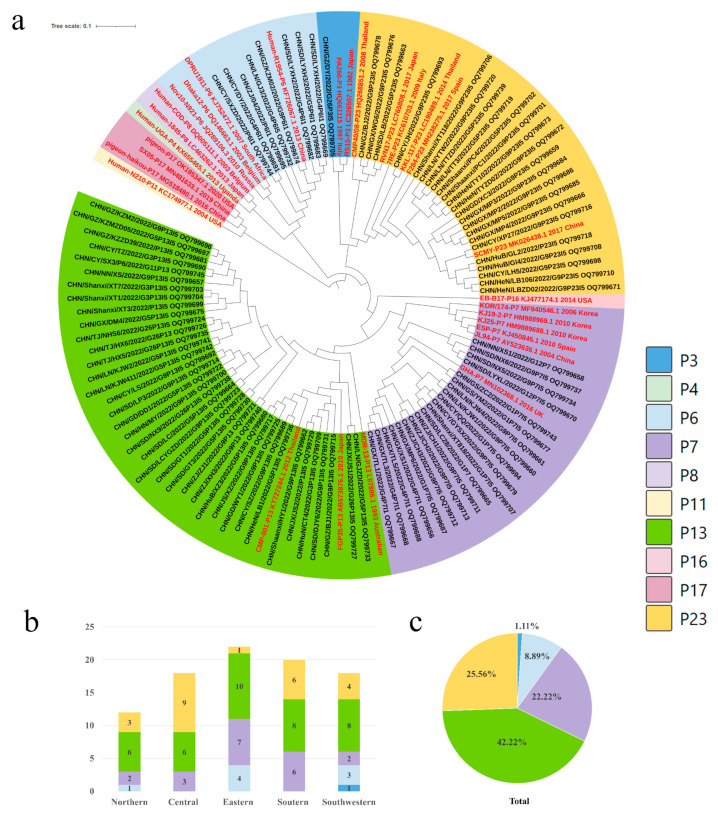
(**a**) Genotype distribution and genetic and phylogenetic analysis of the VP4 gene in different regions from diarrheal pigs. The trees of the VP4 gene were created via neighbor-joining analysis using MEGA 11 software with 1000 bootstrap replicates. Sequences of the various P genotypes of reference RVA strains were obtained from GenBank. The reference sequence was shown in red. (**b**) The genotype proportions of the VP4 gene at the farm level in different regions in 2022. (**c**) Proportional distribution of genotypes in the VP4 gene in 2022.

**Figure 4 viruses-16-00453-f004:**
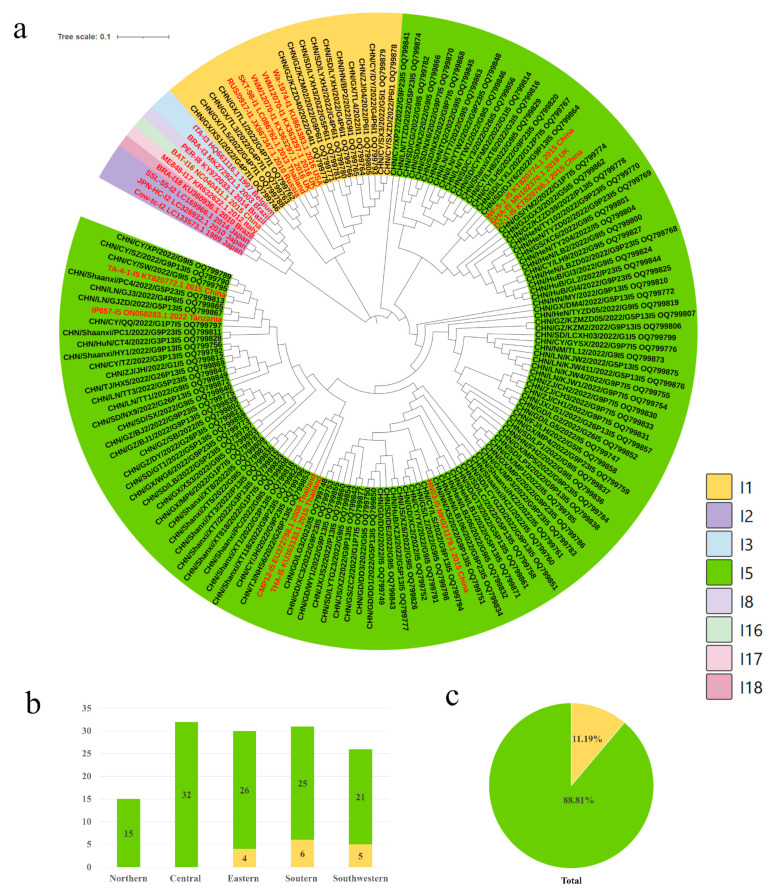
(**a**) Genotype distribution and genetic and phylogenetic analysis of the VP6 gene in different regions from diarrheal pigs. The trees of the VP6 gene were created via neighbor-joining analysis using MEGA 11 software with 1000 bootstrap replicates. Sequences of the various I genotypes of reference RVA strains were obtained from GenBank. The reference sequence was shown in red. (**b**) The genotype proportions of the VP6 gene at the farm level in different regions in 2022. (**c**) Proportional distribution of genotypes in the VP6 gene in 2022.

**Figure 5 viruses-16-00453-f005:**
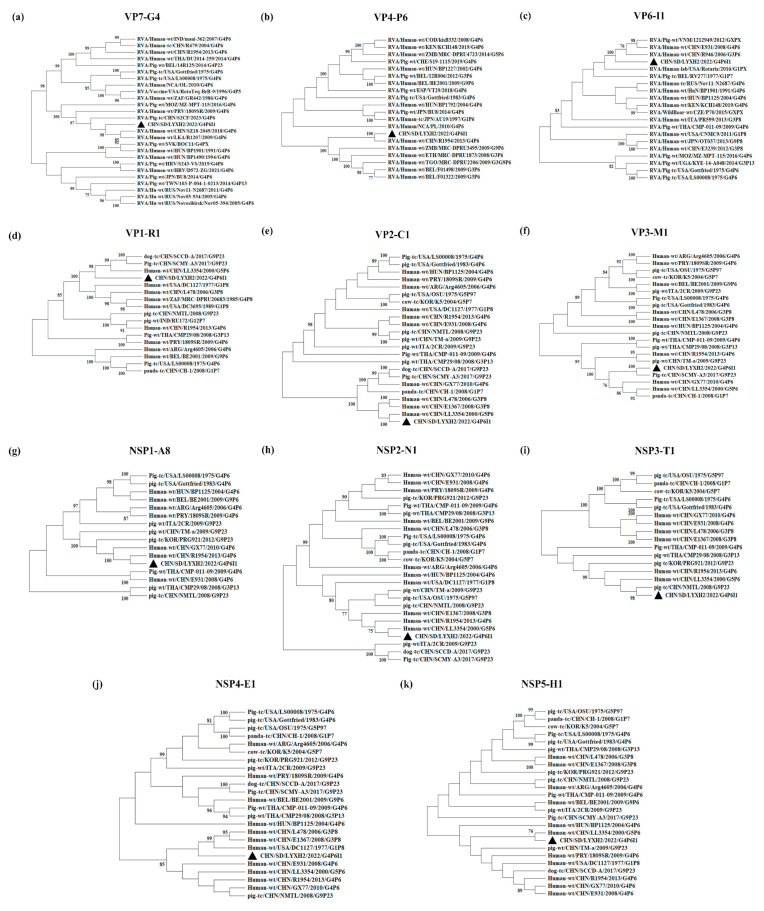
Phylogenetic dendrogram based on the nucleotide sequences of VP7 (**a**), VP4 (**b**), VP6 (**c**), VP1 (**d**), VP2 (**e**), VP3 (**f**), NSP1 (**g**), NSP2 (**h**), NSP3 (**i**), NSP4 (**j**) and NSP5 (**k**) from LYXH2 and reference strains. The trees were created via neighbor-joining analysis using MEGA 11 software with 1000 bootstrap replicates. ▲ The sequences in this study.

**Table 1 viruses-16-00453-t001:** PoRVA positive samples and farms in China, 2022.

China Region	Province	No. of Tested Samples	No. of Positive Samples	Positive Rate at the Sample Level (%)	No. of Tested Farms	No. of Positive Farms	Positive Rate at the Farm Level (%)
Northern	Hebei	627	250	39.87	6	5	83.33
	Heilongjiang	155	135	87.10	2	2	100.00
	Liaoning	1490	891	59.80	21	19	90.48
	Inner Mongolia	915	558	60.98	11	10	90.91
	Tianjin	182	97	53.30	10	7	70.00
	Total	3369	1931	57.32	50	43	86.00
Central	Gansu	1585	819	51.67	14	13	92.86
	Henan	1440	843	58.54	11	10	90.91
	Shanxi	1411	1217	86.25	4	4	100.00
	Shaanxi	1894	567	29.94	9	8	88.89
	Hubei	1682	1351	80.32	11	10	90.91
	Total	8012	4797	59.87	49	45	91.84
Eastern	Anhui	894	475	53.13	5	4	80.00
	Shandong	1302	452	34.72	17	13	76.47
	Jiangsu	1212	538	44.39	10	9	90.00
	Zhejiang	1310	563	42.98	12	11	91.67
	Total	4718	2028	42.98	44	37	84.09
Southern	Fujian	329	208	63.22	4	3	75.00
	Guangdong	1333	735	55.14	7	6	85.71
	Guangxi	1645	816	49.60	16	13	81.25
	Hainan	264	97	36.74	5	4	80.00
	Hunan	1548	466	30.10	11	10	90.91
	Jiangxi	1597	728	45.59	12	11	91.67
	Total	6716	3050	45.41	55	47	85.45
Southwestern	Sichuan	1746	677	38.77	21	18	85.71
	Guizhou	1100	649	59.00	7	6	85.71
	Yunnan	107	48	44.86	4	3	75.00
	Total	2953	1374	46.53	32	27	84.38
Total		25,768	13,180	51.15	230	199	86.52

**Table 2 viruses-16-00453-t002:** PoRVA genotype combination analysis.

	P3	P6	P7	P[13]	P[23]	
G1						I1
			5			I5
G3						I1
				4		I5
G4		3	4			I1
		1				I5
G5		1				I1
				11	4	I5
G9		1				I1
			8	12	18	I5
G11				1		I1
						I5
G12						I1
			1			I5
G26						I1
	1			4		I5

**Table 3 viruses-16-00453-t003:** Comparison of CHN/SD/LYXH2/2022/G4P6I genes with those of human and animal rotaviruses.

Strain	Host	Genotypes of Viral Protein Genes and Nucleotide Sequence Identities (% *) to LYXH2																					
		VP7		VP4		VP6		VP1		VP2		VP3		NSP1		NSP2		NSP3		NSP4		NSP5	
LYXH2	Pig	G4	- ^#^	P6	-	I1	-	R1	-	C1	-	M1	-	A8	-	N1	-	T1	-	E1	-	H1	-
ET8B/2015	Pig	G5	72.4	P[13]	63.7	I5	84.2	R1	85.8	C1	86.9	M1	86.5	A8	89.7	N1	89.6	T7	83.6	E1	90.8	H1	98.3
HeNNY	Pig	G4	85.2	P[23]	71.1	I5	82.8	R1	90.4	C1	86.9	M1	97.4	A8	81.2	N1	89.9	T1	93.6	E1	95.3	H1	96.1
LNCY	Pig	G3	73.6	P[13]	68.2	I5	83.5	R1	86.0	C1	87.5	M1	97.0	A8	81.4	N1	89.6	T1	94.2	E1	98.7	H1	97.6
CMP-011	Pig	G4	84.0	P6	93.9	I1	90.8	R1	86.8	C1	86.6	M1	87.8	A8	85.4	N1	89.5	T1	91.4	E1	89.2	H1	96.6
SD-1	Pig	G9	74.9	P[23]	71.4	I5	83.6	R1	85.7	C1	95.5	M1	95.3	A8	81.9	N1	90.3	T1	94.0	E1	94.5	H1	97.3
CN127	Pig	G12	72.1	P7	68.3	I1	94.6	R1	85.9	C1	86.3	M1	95.1	A8	88.7	N1	87.8	T1	88.0	E1	96.5	H1	97.0
SCLSHL	Pig	G9	74.6	P[23]	71.9	I1	-- ^&^	R1	88.2	C1	86.5	M1	95.6	A8	95.8	N1	93.0	T1	94.1	E1	92.5	H1	97.8
SCMY	Pig	G9	74.4	P[23]	72.2	I5	83.5	R1	92.8	C1	86.6	M1	97.5	A8	82.8	N1	87.6	T1	92.3	E1	87.9	H1	97.0
LS00008	Pig	G4	83.6	P6	84.5	I1	90.8	R1	86.1	C1	86.5	M1	86.6	A8	87.2	N1	90.4	T1	87.6	E1	89.5	H1	97.0
GX54	Human	G4	84.7	P6	96.3	I1	94.6	R1	88.1	C1	86.9	M1	96.4	A8	95.3	N1	89.4	T1	88.1	E1	93.0	H1	98.0
SCLS-R3	Human	G3	74.3	P[13]	69.9	I5	81.3	R1	88.6	C1	86.6	M1	93.4	A8	95.7	N1	88.6	T7	86.6	E1	93.2	H1	98.0
E931	Human	G4	84.3	P6	96.5	I1	96.5	R1	88.6	C1	87.4	M1	85.0	A8	86.1	N1	89.9	T1	88.3	E1	93.7	H1	98.0
R946	Human	G3	72.9	P6	94.9	I1	96.7	R1	88.2	C1	86.9	M1	84.8	A1	77.1	N1	94.5	T1	94.4	E1	92.1	H1	97.0
R1954	Human	G4	84.8	P6	96.7	I1	96.6	R1	88.4	C1	86.5	M1	95.8	A8	95.8	N1	94.7	T1	94.6	E1	93.0	H1	97.5
R1207	Human	G4	96.0	P6	94.7	I1	96.6	R1	86.1	C1	87.2	M1	95.1	A1	77.0	N1	91.6	T1	87.6	E1	89.0	H1	96.8
LL3354	Human	G5	73.1	P6	95.5	I5	83.7	R1	93.8	C1	95.5	M1	95.8	A1	77.0	N1	96.0	T1	94.4	E1	93.7	H1	98.5
R479	Human	G4	84.4	P6	95.2	I5	82.9	R1	88.6	C1	86.9	M1	95.3	A1	77.5	N1	88.9	T7	85.5	E1	89.2	H1	98.5
KNA/08979	Simian	G5	72.1	PX	-- ^&^	I5	84.3	R1	86.7	C1	86.7	M1	86.5	A8	89.7	N1	89.2	T7	83.6	E1	90.8	H1	98.3
SCCD-A	Dog	G9	75.2	P[23]	72.2	I5	83.3	R1	92.9	C1	86.5	M1	97.3	A8	80.4	N1	87.4	T7	86.8	E1	87.9	H1	97.8

* The nucleotide (nt) sequence identity value (nt/aa), expressed as a percentage, was calculated based on the complete ORF sequence of each genetic fragment. The distance matrix for nucleotide (nt) sequences was generated using the p-distance algorithm in Mega 7.0. ^#^ Self-homology is 100%, which is meaningless. ^&^ Gene sequences not obtained in the original research.

## Data Availability

All data analyzed during this study are included in this published article. The raw data generated during the current study are available from the corresponding author on reasonable request. The nucleotide sequences were deposited in GenBank under the accession numbers OQ743847-OQ743991 (VP7), OQ799656-OQ799745 (VP4), OQ799746-OQ799879 (VP6), and OQ799880-OQ799887 (LYXH2 VP1-NSP5).

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
