# Peer review of "Recent Molecular Characterization of Porcine Rotaviruses Detected in China and Their Phylogenetic Relationships with Human Rotaviruses"

_viruses, 2024, doi:10.3390/v16030453_

Round 1

Reviewer 1 Report

Comments and Suggestions for Authors

Porcine Rotavirus A (PoRVA) is a gastrointestinal pathogen that causes severe diarrhea in young piglets. Understanding its prevalence and molecular properties is critical for disease prevention.

In 2022, the authors of this manuscript collected 25,768 fecal samples from 230 farms in China. They discovered that 86.52% of the samples tested positive for porcine RVA, with 51.15% showing positive results. Among the VP7 genotypes, G9 was dominant, accounting for 56.55%. The three major VP4 genotypes were P[13] (42.22%), P[23] (25.56%), and P7 (22.22%). VP6 was found in only two genotypes: I5 (88.81%) and I1 (11.19%). The most common RVA genotype observed was G9P[23]I5. Additionally, the researchers identified a distinct PoRVA strain, CHN/SD/LYXH2/2022/G4P6I1, which shared high homology with human RV strains, suggesting genetic re-assortment between human and porcine RV strains.

This study not only updated the prevalence of RVA in China but also provided additional information for the epidemiology of RVA.

However, it is essential to include a description of the coverage of NSP3 qRT-PCR to rule out potential sample selection bias. While the study’s primers or probe match with the majority of known rotavirus NSP3, it is important to note that there are exceptions. The limitations regarding these exceptions should be explained. Additionally, in lines 159 to 162, where it states that the number of VP4 full-length sequenced samples is 90, this significantly fewer number compared to those for VP6 and VP7 could potentially undermine the reliability of the study’s conclusions.

Reviewer 2 Report

Comments and Suggestions for Authors

The authors reported a high prevalence of RVA among porcine farms in 5 regions of China. They also described the main genotypes circulating and reported cross-species transmission.

Major revisions

11) The authors reported a prevalence of RVA in pigs significantly higher than that described by previous studies in the same regions in the last few years. How they can explain this situation?

22) The study describes a high interspecies transmission. The authors should also analyse the origin of reassortant strains. Did recombination events occur?

3) The authors should include more reference sequences from other countries in the phylogenetical analyses to reach a better classification

Minor revisions

4)     The sentence in the introduction (lanes 61-62) is not clear. What do you mean that there is a large number of genotypes determined by the VP6 gene?

5)     The authors should indicate and briefly describe the sequencing method used in the study. A Sanger approach was used?

6)     The authors should provide more information on pig management. The farming type? They are industrial or backyard raising?

7)     How it was conducted VP7, VP6 and VP4 genotyping? A genotyping tool was employed? They should include such information in the method section.

8)     Figure 2-3-4.  the authors should include the geographical origin of the reference sequences and specify which are reference sequences in the caption of figures.

Round 2

Reviewer 2 Report

Comments and Suggestions for Authors

In the revised version of Supplementary Materials, Supplementary S3 is missing. The authors have to include it.

Author Response

Thank you for your suggestion. In the compressed file of the modified file that was uploaded a few days ago, I uploaded the supplementary table 3. Herein, I have separately uploaded the supplementary table 3 in the attachment. Please check it. Thanks again.
